# Heterologous Expression of *Platycodon grandiflorus PgF3′5′H* Modifies Flower Color Pigmentation in Tobacco

**DOI:** 10.3390/genes14101920

**Published:** 2023-10-09

**Authors:** Lulin Ma, Wenjie Jia, Qing Duan, Wenwen Du, Xiang Li, Guangfen Cui, Xiangning Wang, Jihua Wang

**Affiliations:** Flower Research Institute of Yunnan Academy of Agricultural Sciences, Key Lab of Yunnan Flower Breeding, National Engineering Research Center For Ornamental Horticulture, Kunming 650205, China; malulin417@163.com (L.M.); leexiang93@163.com (X.L.);

**Keywords:** *PgF3′5′H*, transgenic tobacco, over-expression, delphinidin-based anthocyanin, blue flower color

## Abstract

Flavonoid-3′,5′-hydroxylase (F3′5′H) is the key enzyme for the biosynthesis of delphinidin-based anthocyanins, which are generally required for purple or blue flowers. Previously, we isolated a full-length cDNA of *PgF3′5′H* from *Platycodon grandiflorus*, which shared the highest homology with *Campanula medium F3′5′H*. In this study, *PgF3′5′H* was subcloned into a plant over-expression vector and transformed into tobacco via *Agrobacterium tumefaciens* to investigate its catalytic function. Positive transgenic tobacco T_0_ plants were obtained by hygromycin resistance screening and PCR detection. *PgF3′5′H* showed a higher expression level in all *PgF3′5′H* transgenic tobacco plants than in control plants. Under the drive of the cauliflower mosaic virus (CaMV) 35S promoter, the over-expressed *PgF3′5′H* produced dihydromyricetin (DHM) and some new anthocyanin pigments (including delphinidin, petunidin, peonidin, and malvidin derivatives), and increased dihydrokaempferol (DHK), taxifolin, tridactyl, cyanidin derivatives, and pelargonidin derivatives in *PgF3′5′H* transgenic tobacco plants by ultra-performance liquid chromatography–tandem mass spectrometry (UPLC-MS/MS) analysis, resulting in a dramatic color alteration from light pink to magenta. These results indicate that *PgF3′5′H* products have F3′5′H enzyme activity. In addition, *PgF3′5′H* transfer alters flavonoid pigment synthesis and accumulation in tobacco. Thus, *PgF3′5′H* may be considered a candidate gene for gene engineering to enhance anthocyanin accumulation and the molecular breeding project for blue flowers.

## 1. Introduction

Flower color not only plays a crucial role in attracting pollinators to complete sexual hybridization but also has a significant ornamental character for humans [1,2,3]. Flower color formation is determined primarily by three kinds of pigments: flavonoids, carotenoids, and betalains. Among them, flavonoids confer a diverse color range from yellow to blue. In particular, anthocyanins, a colored class of flavonoids, contribute to orange, red, magenta, violet, and blue flower colors [1,4,5]. Anthocyanidins include six major classes: pelargonidin, cyanidin, peonidin, delphinidin, petunidin, and malvidin [4,6]. Peonidin is a methylated derivative of cyanidin; petunidin and malvidin are methylated derivatives of delphinidin [6,7]. Pelargonidin mainly produces orange or brick red flowers, cyanidin and peonidin produce pink or red flowers, while delphinidin, petunidin, and malvidin tend to yield violet or blue flowers [6,7,8,9,10,11]. Several factors further affect the final flower color, including the chemical structure of anthocyanins, co-pigments, metal ions, and vacuole pH [9]. The chemical structure of anthocyanins greatly influences anthocyanin color and, thus, flower color [1,5,8,12]. For instance, with an increase in the number of hydroxyl groups on the B-ring, the amount of delphinidin-based anthocyanins increases, resulting in a bluer flower color [1,6,8,13,14,15,16].

Creating distinct flower colors in ornamental plants has long been a major goal of breeding. Breeders have cultivated large varieties of flower colors using traditional breeding techniques. However, flower colors are still lacking in many ornamental plants due to the limited gene pool of a single species. For example, the most popular cut flower crops, such as roses (*Rosa* spp.), carnations (*Dianthus caryophyllus*), chrysanthemums (*Chrysanthemum morifolium*), and lilies (*Lilium* spp.), do not have blue-flowering varieties due to a deficiency in flavonoid-3′,5′-hydroxylase (F3′5′H), the key enzyme responsible for the synthesis of delphinidin-based anthocyanins [1,2,6,14,17,18].

F3′5′H is a cytochrome P450 (Cyt P450) enzyme, which catalyzes the 3′,5′-hydroxylation of dihydrokaempferol (DHK) to form dihydromyricetin (DHM), a precursor of delphinidin for violet or blue pigments [8,19,20]. As a result, the *F3′5′H* gene is often referred to as the “blue gene” [12,21,22], and it has attracted considerable attention as a crucial molecular tool for generating blue flower colors through genetic engineering techniques [6]. Since the first *F3′5′H* was isolated from *Petunia hybrida* [23], to date, the homologous genes of *F3′5′H* have been isolated from many plants, including *C. medium* [24], *Vinca major* [25], *Verbena hybrida* [26], *Phalaenopsis* [27,28], *Glycine soja* [29], *Cyclamen persicum* [30], *Dendrobium moniliforme* [31], *Antirrhinum kelloggii* [8], *Epimedium sagittatum* [32], *Pericallis hybrida* [22], *Camellia sinensis* [12], and *Aconitum carmichaelii* [33,34]. By heterologous over-expression, some *F3′5′Hs* have been used to increase the accumulation of delphinidin-based anthocyanins for producing a blue color in some flower crops, such as roses [1,2,6,18], carnations [2,6,17,18] and chrysanthemums [7,14].

*P. grandiflorus*, a perennial herb, is the only species of the genus *Platycodon* and belongs to the Campanulaceae family. It has been widely used as a traditional herbal medicine and food resource in Northeast Asian countries, such as China, Japan, and Korea [35,36,37]. In addition, *P. grandiflorus* can also be used as an ornamental plant, due to its bell-shaped and bright blue-colored flowers. Previously, we obtained a full-length cDNA sequence of *F3′5′H* (GenBank accession JQ403611, named *PgF3′5′H*) from *P. grandiflorus* flowers. An NCBI BLAST (Basic Local Alignment Search Tool) (https://blast.ncbi.nlm.nih.gov/Blast.cgi, accessed on 5 January 2013) performed with the protein sequence of *PgF3′5′H* showed that the closest sequence was the *C*. *medium F3′5′H* (GenBank accession D14590), and gene expression analysis revealed that *PgF3′5′H* was highly expressed in the blue buds, but absent from the leaves [38]. It is speculated that *PgF3′5′H* may play a significant role in the biosynthesis of blue anthocyanins in *P. grandiflorus* flowers [38]. It is unclear, however, whether *PgF3′5′H* really has in vivo F3′5′H enzyme activity to synthesize delphinidin-based anthocyanins for blue or purple petals in transgenic plants. In the present study, we report that over-expression of *PgF3′5′H* exhibited a deeper flower color in transgenic tobacco plants than in control plants, due to the production of delphinidin-based anthocyanins. It suggests that *PgF3′5′H* may be a valuable genetic resource for the molecular breeding of blue-flowered ornamental species through genetic engineering.

## 2. Materials and Methods

### 2.1. Materials

Tobacco (*Nicotiana tabacum* L. cv. K326) was used as a host for the genetic transformation of *PgF3′5′H*. Tobacco K326 seeds were provided by the Yunnan Academy of Tobacco Agricultural Sciences. The plant expression vector pCAMBIA1301-KY, which was designed to over-express the *PgF3′5′H* gene under the cauliflower mosaic virus (CaMV) 35S promoter, and the *A. tumefaciens* strain LBA4404 were offered by Shanghai Jin Chao Technology Development Co., Ltd. (Shanghai, China). The plasmid of pMD18T_*PgF3′5′H* was provided by the Yunnan Flower Breeding Key Laboratory, Flower Research Institute, Yunnan Academy of Agricultural Sciences.

### 2.2. Construction of Plant Expression Vector and Tobacco Transformation

After sterilization in 75% alcohol for 1 min and then in 15% H_2_O_2_ for 15 min, tobacco seeds were grown aseptically on MS medium, and cultured in a light incubator at 28 °C for 15 days.

Using the plasmids of pMD18T_*PgF3′5′H* as PCR reaction templates, the full ORF of *PgF3′5′H* was amplified by gene-specific primers F3′5′H-XbaI: 5′-TGC TCT AGA GCA ATG GCT ATA GAC ACA ATT AT-3′ and F3′5′H-BamHI: 5′-CGC GGA TCC GCG CTA GGC AGT GTA AGC ACT TG-3′, containing the *Xba*I restriction sites (the underlined sequence in primer F3′5′H-XbaI) at the 5′ end and *Bam*HI restriction sites (the underlined sequence in primer F3′5′H-BamHI) at the 3′ end, respectively. The PCR amplification fragments were digested by *Xba*I and *Bam*HI and ligated into the plant expression vector (also digested with *Xba*I and *Bam*HI) to form the pCAMBIA1301-KY-*PgF3′5′H* recombinant vector (Figure 1).

The recombinant vector was transformed into tobacco K326 plants via *A. tumefaciens* LBA4404 with the leaf disc method [39], and pCAMBIA1301-KY was transformed as an empty vector control. Transgenic T_0_ generation tobacco plants were screened through the evaluation of hygromycin (25 mg/L) resistance and PCR detection (the detection primers for *PgF3′5′H* in transgenic tobacco were 35S-F: 5′-GAC GCA CAA TCC CAC TAT CC-3′ and F3′5′H-BamHI; the primers for empty vector control were 35S-F and GUS-R: 5′-GAA AAG GGT CCT AAC CAA GA-3′, respectively), and then transplanted into the greenhouse for later observation and analysis. The transgenic tobacco plants were transplanted into a greenhouse under temperature conditions of 25 °C/20 °C day/night, with 16 h of light and 8 h of darkness and a relative humidity of 55–75%.

During the flowering stages, the flower color phenotypes of *PgF3′5′H* transgenic T_0_ generation plants were compared with empty vector plants and non-transformed wild-type K326 plants. The fresh full-bloom flowers of several transgenic T_0_ generation tobacco lines with obvious magenta color changes were used as samples for the next analysis of gene expression and metabolic profiling. Wild-type K326 flowers with 3–4 biological replicates were used as non-transformed controls for result accuracy. For each experimental sample, 20 flowers with the same lines and conditions were mixed together and stored at −80 °C until further use [40].

### 2.3. qRT-PCR Analysis for PgF3′5′H Expression in Transformed Tobacco

The modified CTAB-based method was used for total RNA extraction from tobacco flowers [41]. RNA quality and integrity were detected using a SMA4000 UV-Vis spectrophotometer (Merinton, Beijing, China) and 1% agarose gel electrophoresis, respectively. The first-strand cDNA was synthesized using MonScript™ RTIII All-in-One Mix (Monad, Suzhou, China) following the manufacturer’s instructions. The cDNA was diluted 10-fold for quantitative real-time PCR (qRT-PCR) analysis.

qRT-PCR was carried out using the QuantiNova SYBR Green PCR Kit (QIAGEN, Dusseldorf, Germany) according to the instructions on Applied Biosystems™ 7500 Real-Time PCR System. Gene-specific primers F3′5′H-F: 5′-GGG GCT AGA ATG GGA ATC GG-3′ and F3′5′H-R: 5′-AGC CTT GGA GTA ACA CTG GC-3′ were designed to analyze *PgF3′5′H* expression. For qRT-PCR normalization, a tobacco *actin* gene (the primers were ACT-F: 5′-CTG AGG TCC TTT TCC AAC CA-3′ and ACT-R: 5′-TAC CCG GGA ACA TGG TAG AG-3′) was used as the reference gene [42]. In the PCR reactions, the following conditions were applied: 95 °C for 2 min, 40 cycles of 95 °C for 5 s, and 60 °C for 30 s. For each sample, three technical replicates were conducted. The relative expression between the positive *PgF3′5′H* transgenic T_0_ lines and the wild-type K326 tobacco plants (which served as the control with three biological replicates) was determined using the 2^−ΔΔCT^ method [43]. qRT-PCR analysis was performed by Norminkoda Biotechnology Co., Ltd. (Wuhan, China).

### 2.4. Extraction and Analysis of Anthocyanins in Transformed Tobacco

Color changes in tobacco flowers are primarily caused by differences in anthocyanin classes and contents [12,22,24,40]. To determine anthocyanins in transgenic tobacco flowers, 50 mg of freeze-dried tobacco petals were ground into a powder and extracted with 0.5 mL methanol/water/hydrochloric acid (500:500:1, by volume). After centrifugation under 12,000× *g* for 3 min at 4 °C, the upper layer solutions were collected and filtered through a membrane filter (0.22 μm, Anpel) for ultra-performance liquid chromatography–tandem mass spectrometry (UPLC-MS/MS) analysis.

The UPLC-MS/MS analysis of anthocyanin compounds was based primarily on the detailed standard procedure described previously [44]. To determine the substrate specificity for PgF3′5′H, nine flavonoids (Table 1), which can be directly catalyzed or synthesized by F3′5′H [12,45,46], were chosen as standards for further testing. The detection and analysis of anthocyanins/flavonoids was performed on the AB Sciex QTRAP 6500 LC-MS/MS platform by Metware Biotechnology Co., Ltd. (Wuhan, China).

Hierarchical cluster analysis (HCA) results of samples and metabolites were presented as heatmaps using *R* software. The variable importance in projection (VIP) value was calculated using partial least squares discriminant analysis (PLS-DA). Metabolites were considered differentially changed between two groups if the VIP ≥ 1, and fold change ≥2 or ≤0.5 [40].

## 3. Results

### 3.1. Phenotype Comparison of Transgenic T_0_ Generation Tobaccos

To investigate the function of over-expressed *PgF3′5′H* on anthocyanin biosynthesis, the binary vector of pCAMBIA1301-KY-*PgF3′5′H* and the empty pCAMBIA1301-KY vector were transformed into tobacco via *A. tumefaciens*, respectively. A total of 13 independent positive transgenic T_0_ generation tobacco plants of *PgF3′5′H* and 10 control transgenic T_0_ tobacco plants of the empty vector were obtained, respectively. The flowers of the *PgF3′5′H* transgenic plants exhibited deeper colors than the controls, ranging from deep pink to magenta. Among all the lines, the W-58 showed the deepest color, magenta during full bloom and bluing during the flower senescence period. However, the flower color of all empty vector transgenic control plants was identical to that of non-transformed K326 plants (Figure 2). Thus, wild-type K326 was chosen as a control to facilitate multiple biological replicates for subsequent qRT-PCR analysis and UPLC-MS/MS detection.

### 3.2. Heterologous Expression of PgF3′5′H in Transformed Tobacco

With tobacco *actin* used as a reference gene, *PgF3′5′H* transcripts were detected in several transgenic T_0_ generation lines with obvious flower color changes by qRT-PCR. The qRT-PCR analysis indicated that *PgF3′5′H* gene expression was significantly higher in all *PgF3′5′H* transgenic T_0_ generation lines than in wild-type K326 plants. *PgF3′5′H* showed an extremely high expression level in transgenic T_0_ generation line W-58 (approximately 7500 times higher than that in wild-type K326 plants), which corresponds to its magenta flower color (Figure 3).

### 3.3. Comparative Analysis of Anthocyanins in Transformed and Wild-Type Tobacco

To identify the anthocyanin components in the *PgF3′5′H* transgenic tobacco flowers, a widely targeted UPLC-MS/MS metabolomics analysis was carried out based on the Metware database, constructed with 108 standard products. A total of twenty-seven anthocyanins, seven flavonoids, and one anthocyanidin were identified in the flowers of the *PgF3′5′H* transgenic and wild-type K326 tobacco plants (Appendix A). Among these pigments, one anthocyanidin (delphinidin) and four anthocyanins (including delphinidin-3-*O*-rutinoside, petunidin-3-*O*-rutinoside, peonidin-3-*O*-glucoside, and malvidin-3-*O*-rutinoside) were detected in the flowers of the transgenic lines, but not in wild-type K326 plants, as shown in the heatmaps (Figure 4). Meanwhile, five differentially accumulated metabolites (DAMs) in flowers between the *PgF3′5′H* transgenic lines and wild-type K326 plants were selected based on VIP ≥ 1 and fold change ≥2, and they were all up-accumulated in *PgF3′5′H* transgenic line flowers (Table 2).

Nine flavonoids were selected as standards for determining the substrate specificity of PgF3′5′H and further confirming the DAM results. According to the UPLC-MS/MS analysis, tricetin (5,7,3′,4′,5′-pentahydroxyflavone) was undetected in the flowers of all *PgF3′5′H* transgenic lines and wild-type K326 plants, while DHM was detected in the flowers of *PgF3′5′H* transgenic plants but not in the wild-type plants (Table 1). Also, three flavonoids, including luteolin, DHK, and dihydroquercetin (DHQ), were up-accumulated in *PgF3′5′H* transgenic lines compared with wild-type plants (Table 1). While the difference in eriodictyol content was deemed insignificant between the two groups due to VIP (0.997) < 1, its content in all transgenic line flowers was much higher than that in wild-type K326 tobacco (Table 1).

## 4. Discussion

F3′5′H and flavonoid 3′-hydroxylase (F3′H) play a crucial role in flower color by introducing hydroxyl groups at the 3′ and 5′ or 3′ positions to determine the B-ring hydroxylation pattern of anthocyanins, respectively [3,6,8,18,47]. In particular, F3′5′H is necessary for the biosynthesis of delphinidin-based anthocyanins, which confer violet or blue coloration on most plant flowers [6,8,18].

In this study, a *P. grandiflorus PgF3′5′H* gene was subcloned into a modified plant expression vector pCAMBIA1301 and expressed under the control of the CaMV35S promoter in K326 tobacco with pink flowers. *PgF3′5′H* expression was detected by qRT-PCR in several transgenic lines with obvious magenta color changes in flowers. *PgF3′5′H* showed more than 2000-fold higher expression levels in all *PgF3′5′H* transgenic tobacco plants than in wild-type K326 plants (Figure 3). All transgenic lines, however, failed to produce true blue flowers as desired, despite the high expression levels of *PgF3′5′H*, even though the maximum expression level (transgenic line W-58) was more than 7500 times higher than in wild-type K326 plants (Figure 3). Our result was also consistent with color modification results in tobacco flowers after the over-expression of *F3’5’H* genes from other plants, such as the *P. hybrida F3′5′H* gene *AK14* [48], the *Eustoma russellianum F3′5′H* gene *TG1* [48], the *C. medium F3′5′H* gene *Ka1* [24], the *P. hybrida F3′5′H* gene *PCFH* [22], the *C. sinensis CsF3′5′H* [12], and the *A. carmichaelii AcF3′5′H* [34], etc. In particular, compared with other *F3′5′H* genes, *C. medium Ka1* resulted in 99% of delphinidin-based anthocyanins out of total anthocyanidins, but did not produce a blue color in tobacco flowers [24]. The reason is relatively complex. On the one hand, the anthocyanin biosynthetic pathway is a complex metabolic network co-regulated by multiple genes in plants [7,12]. The over-expression of exogenous *F3′5′H* genes is insufficient to efficiently accumulate delphinidin-based anthocyanins, due to some endogenous genes, including *F3′H*, dihydroflavonol 4-reductase (*DFR*), and flavonol synthase (*FLS*), often competing against the introduced *F3′5′H* [6,18]. To avoid competing with F3′5′H, these competing genes should be regulated down [18]. Moreover, the expression of some genes, which can promote the accumulation of delphinidin-based anthocyanins in cooperation with *F3′5′H*, such as *DFR*, glucosyltransferase (*GT*), and anthocyanin acyltransferase (*AT*) genes, should be increased [2,14,49]. For example, to achieve a more exclusive and dominant accumulation of delphinidin to obtain blue rose flowers, researchers down-regulated the endogenous DFR and over-expressed both a *Viola wittrockiana F3′5′H* and an *Iris hollandica DFR* genes, which were introduced into the pink flower rose cultivars [1,18,49]. Blue-colored flowers in transgenic chrysanthemums were dependent on both the co-expression of the *C. medium F3′5′H* (*CamF3′5′H*) gene and *Clitoria ternatea* UDP-glucose: anthocyanin 3′,5′-*O*-glucosyltransferase (*CtA3′5′GT*) gene and the resulting accumulation of 3′,5′-glucosylated delphinidin-based anthocyanins [14,50]. On the other hand, these results indicate that blue flowers are not necessarily produced solely by anthocyanins [12,24]. It is necessary to accumulate delphinidin-based anthocyanins to produce a purple or violet color, but it is not enough to produce true blue flowers [14]. Apart from anthocyanins (including their structure, type, concentration, and localization), the co-existing compounds (co-pigments), metal ion type and concentration, pH of vacuoles, and shapes of surface cell can all influence the formation of the final flower color [9]. In our study, however, we observed a certain blue color change during the flower senescence period in *PgF3′5′H* transgenic tobacco plants (Figure 2). The result was similar to that of *PCFH* heterologous expression in tobacco [22]. It is possible that the change in color was due to a shift in vacuole pH [22].

Based on the results of metabolomic analysis by UPLC-MS/MS, five new pigments, including delphinidin, delphinidin-3-*O*-rutinoside, petunidin-3-*O*-rutinoside, peonidin-3-*O*-glucoside, and malvidin-3-*O*-rutinoside (Figure 4), as well as one new flavonoid DHM, were detected in the flowers of the *PgF3′5′H* transgenic tobacco plants, but not in wild-type K326 plants (Table 1). Except for peonidin-3-*O*-glucoside, the other four pigments are all delphinidin-based anthocyanins, which produce a violet or blue coloration [6,8,9,10,11]. DHM, a 3′,5′-hydroxylasesis product catalyzed by F3′5′H, is the precursor of delphinidin [8,19,20]. Meanwhile, compared with wild-type K326 plants, four anthocyanins (Pel-3-*O*-glu, Pel-3-*O*-rut, Pel-3-*O*-(6-*O*-mal)-glu and Peo-3-*O*-rut) (Table 2) and three flavonoids (luteolin, DHQ and DHK) (Table 1 and Table 2) were up-accumulated in *PgF3′5′H* transgenic tobacco flowers. Also, cyanidin-3-*O*-rutinoside showed higher levels in most transgenic plants than in wild-type plants, but was labelled ‘Insig’ because of its fold change (1.994) <2 (Appendix A). We speculate that the magenta color may have been caused by the combination of the new pigments and the up-accumulated pigments in *PgF3′5′H* transgenic tobacco flowers. However, these up-accumulated pigments and the new peonidin-3-*O*-glucoside may also inhibit the development of blue flowers, since they are primarily responsible for the production of orange, brick red, pink, and red flower colors [1,2,6,7,18]. Through the co-modification engineering of multiple genes, previous researchers have generated blue-hued flowers in roses and chrysanthemums by reducing the ratio of cyanidin and pelargonidin and increasing the accumulation of delphinidin [1,6,14,18,50].

F3’5’Hs have been shown to exhibit a broad substrate specificity through expression in yeasts and plants. They can hydroxylate naringenin to eriodictyol, naringenin and eriodictyol to tricetin, DHK into DHQ, DHK and DHQ to DHM, and apigenin to luteolin, and so on [12,23,45,46,47,51]. It is possible, however, that some different F3′5′Hs have different substrate specificities. For example, *Solanum lycopersicum* F3′5′H (CYP75A31) uses luteolin, naringenin, eriodictyol, DHK, DHQ, kaempferol, quercetin, and liquiritigenin as substrates [46]. *Gentiana triflora* F3′5′H catalyzes the hydroxylation of naringenin to eriodictyol, eriodictyol to tricetin, DHK to DHQ, DHQ to DHM and apigenin to luteolin, but cannot catalyze kaempferol [45]. And *P. hybrida F3′5′H* (*Hf1*) can accept apigenin as a substrate in the *Escherichia coli* expression system [51]; however, the other *P. hybrida F3′5′H* gene *Hf2* expressed in yeast cannot hydroxylate apigenin [45]. In the present study, eriodictyol was significantly higher in all *PgF3′5′H* transgenic tobacco plants than in wild-type K326 plants, although its VIP (0.097) is less than 1, while tricetin was absent in all tobacco plants (Table 1). Previous studies have reported that, when using naringenin as a substrate, only eriodictyol was detected and no tricetin was detected for *C. sinensis CsF3′5′H* expression in tea [12] and for *G. triflora F3′5′H* heterologous expression in yeast [45], respectively. We did not know whether the tricetin might be completely transformed into other final products, or if the eriodictyol efficiently transformed into DHQ by flavanone 3-hydroxylase (F3H) so that it could not be converted into tricetin. Interestingly, in *PgF3′5′H* transgenic tobacco plants, all dihydroflavonols (DHQ, DHK and DHM) were significantly higher than in wild-type K326 plants (DHM was not detected in all wild-type plants) (Table 1 and Table 2). It has been reported that heterologous *CsF3′5′H* over-expression in tobacco can up-regulate flavonoid pathway genes, including chalcone synthase (CHS), *F3H*, anthocyanidin synthase (*ANS*), anthocyanidin reductase (*ANR*), and UDP-glycose flavonoid glycosyltransferase (*UFGT*) [12]. These up-regulated genes, such as *CHS* and *F3H*, might explain why DHK content was higher in transgenic plants than in wild-type plants in this research. However, it remains to be determined whether *PgF3′5′H* also stimulates these tobacco flavonoid/anthocyanin pathway-related genes, and whether these up-regulated genes together with *PgF3′5′H* promoted or catalyzed up-accumulated dihydroflavonols (DHQ and DHK) and newly generated DHM. PgF3′5′H may also catalyze apigenin into luteolin, as luteolin is up-accumulated in transgenic plants. These results also indicate that *PgF3′5′H* may, like other *F3′5′Hs,* such as *CsF3′5′H* [12], *CYP75A31* [46], *CYP75A8*, and *Hf1* [51], perform not only 3′,5′-hydroxylation, but also 3′-hydroxylation (which is generally completed by *F3′H*) in the flavonoid/anthocyanin pathway. The substrate specificity and preference of PgF3′5′H, however, need to be further determined in vivo or in vitro as well.

## 5. Conclusions

In this study, a *PgF3’5’H* gene from *P. grandiflorus* was transformed into tobacco via *A. tumefaciens*. The expression level of *PgF3’5’H* was shown to be relatively higher in transgenic plants than in controls. Heterologous expression of *PgF3′5′H* resulted in the production of new delphinidin-based anthocyanins and increased the content of cyanidin-based anthocyanins, pelargonidin-based anthocyanins, and flavonoids, and eventually produced a deeper magenta color in transgenic tobacco flowers, compared to the control pink flower. These results show that PgF3′5′H exhibited the hydroxylation activity of the F3′5′H enzyme. It can be considered a candidate gene and applied to the molecular breeding of ornamental plants for blue flowers using gene modification.

## Figures and Tables

**Figure 1 genes-14-01920-f001:**
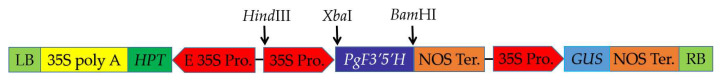
Schematic diagrams of T-DNA region structures of the pCAMBIA1301-KY-*PgF3′5′H* binary vectors constructed for tobacco transformation. LB: left border repeat of T-DNA; 35S ploy A: CaMV 35S polyadenylation signal; *HPT*: hygromycin phosphotransferase gene; E 35S Pro: enhanced CaMV 35S promoter; 35S Pro: CaMV 35S promoter; *PgF3′5′H*: *P*. *grandiflorus* flavonoid-3′,5′-hydroxylase gene containing the restriction sites of *Xba*I and *Bam*HI; NOS Ter.: nopaline synthase gene terminator; *GUS*: *β*-glucuronidase gene; RB: right border repeat of T-DNA.

**Figure 2 genes-14-01920-f002:**
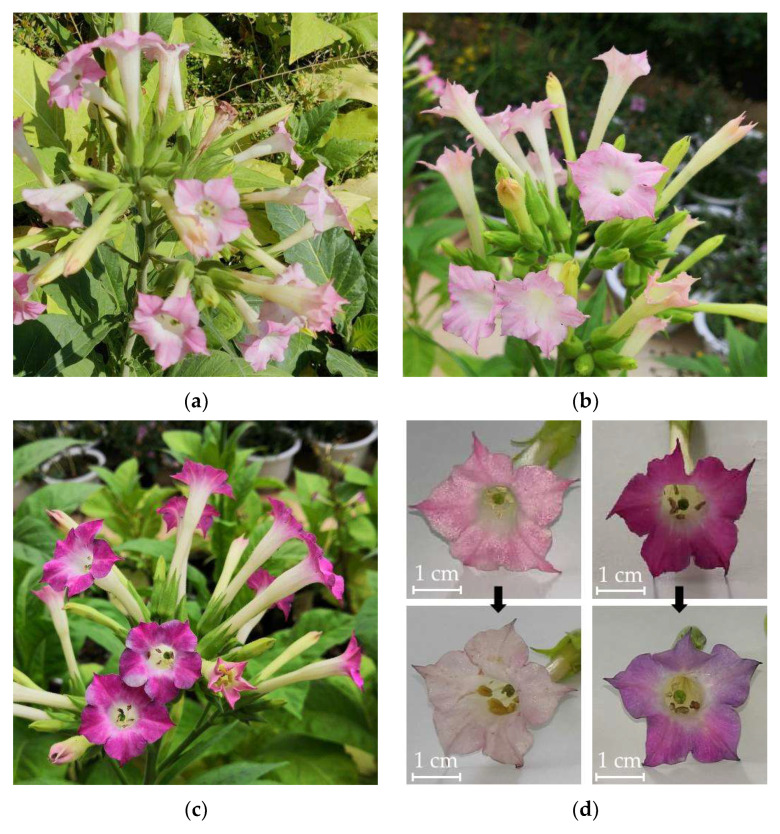
Flower color phenotypes of *PgF3′5′H* transgenic T_0_ generation tobacco. (**a**) Empty vector transgenic plant; (**b**) non-transformed wild-type tobacco K326 plant; (**c**) *PgF3′5′H* transgene line W-58 plant; (**d**) different flower stages of empty vector transgenic control plant (**left**) and *PgF3’5’H* transgene line W-58 plant (**right**). Upper: full-bloom period, lower: flower senescence period.

**Figure 3 genes-14-01920-f003:**
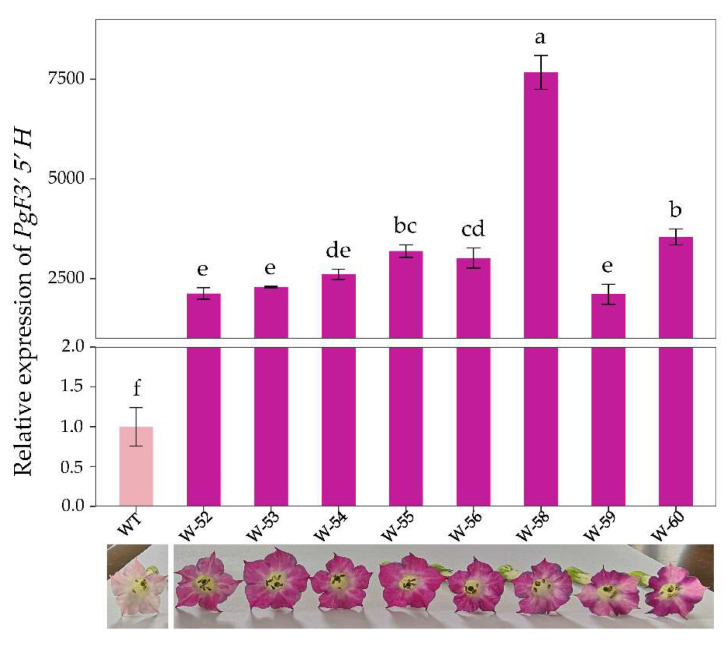
qRT-PCR expression of *PgF3′5′H* and flower color in T_0_ transgenic tobacco plants. Error bars indicate standard deviations and different letters above the bars represent significant differences (*p* < 0.05) according to Tukey’s multiple range test. WT: wild-type K326 tobacco plants; W52-W60: *PgF3′5′H* transgenic T_0_ generation lines.

**Figure 4 genes-14-01920-f004:**
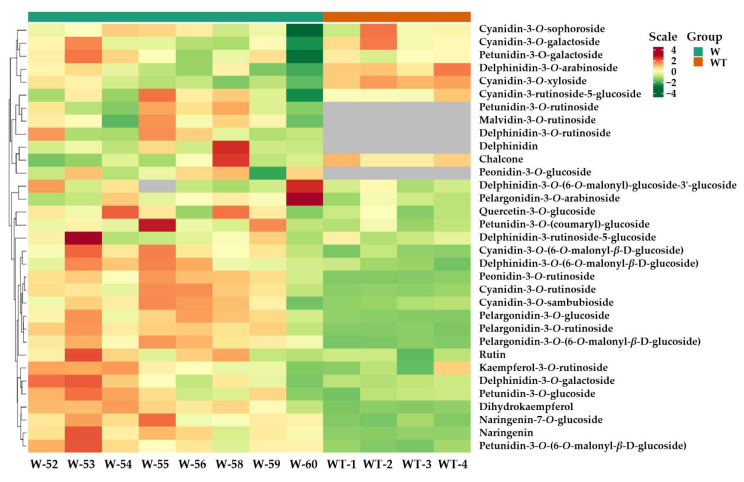
Heatmap visualization of the DAMs identified between *PgF3′5′H* transgenic T_0_ generation lines and wild-type K326 tobacco plants based on 35 pigment metabolite profiles. Color scale from dull red to dark green represents the normalized metabolite contents using row Z-score. The grey colors represent the undetected substances; W52-W60: *PgF3′5′H* transgenic T_0_ generation lines, WT1-WT4: biological replicate plants of wild-type K326 plants.

**Table 1 genes-14-01920-t001:** Detection results of nine flavonoids in the flowers of *PgF3′5′H* transgenic lines and wild-type K326 plants.

Samples	WT-1	WT-2	WT-3	WT-4	W-52	W-53	W-54	W-55	W-56	W-58	W-59	W-60	VIP	Fold_Change	Type
DHM	N.d	N.d	N.d	N.d	0.026	0.019	0.018	0.031	0.016	0.021	0.016	0.018	1.117	Inf	/
Kaempferol	0.072	0.066	0.070	0.067	0.107	0.105	0.108	0.119	0.109	0.088	0.117	0.130	1.062	1.607	Insig
Luteolin	0.029	0.028	0.028	0.028	0.174	0.148	0.134	0.148	0.120	0.156	0.116	0.121	1.122	4.973	Up
DHQ	1.776	1.737	1.746	1.765	69.367	39.973	50.996	55.477	38.780	68.377	35.011	42.298	1.063	28.499	Up
Apigenin	0.073	0.073	0.072	0.076	0.048	0.052	0.216	0.043	0.056	0.040	0.037	0.047	0.070	0.917	Insig
Eriodictyol	0.291	0.278	0.279	0.294	12.239	8.516	6.993	5.814	3.715	8.026	4.906	5.361	0.997	24.346	Insig *
DHK	1.814	1.846	1.831	1.856	4.429	4.008	4.447	3.892	3.757	4.566	3.381	2.671	1.034	2.120	Up
Naringenin	0.861	0.850	0.849	0.868	1.248	1.378	1.251	1.408	1.409	1.034	1.322	1.224	1.081	1.498	Insig
Tricetin	N.d	N.d	N.d	N.d	N.d	N.d	N.d	N.d	N.d	N.d	N.d	N.d	/	/	/

The unit was μg/g, and all values were accurate to 3 decimal places. WT1-WT4: biological replicates of wild-type K326 control tobacco; W52-W60: *PgF3′5′H* T_0_ transgenic generation tobacco lines. N.d: the substance was not detected in this study; Inf: the value tends to infinity; Insig: the difference was insignificant between the two groups; *: There was an insignificant difference in eriodictyol content between the two groups due to the VIP < 1, despite its content in all transgenic line flowers being much higher than that in WT plants.

**Table 2 genes-14-01920-t002:** Up-accumulated pigments in tobacco plants of *PgF3′5′H* transgenic lines.

Samples	WT-1	WT-2	WT-3	WT-4	W-52	W-53	W-54	W-55	W-56	W-58	W-59	W-60	VIP	Fold_Change
Pel-3-*O*-glu	0.043	0.040	0.039	0.039	0.097	0.138	0.080	0.114	0.134	0.121	0.112	0.067	1.225	2.671
Pel-3-*O*-rut	5.606	5.534	5.937	5.182	18.993	22.351	16.427	18.806	19.151	17.274	18.770	9.389	1.252	3.171
Pel-3-*O*-(6-*O*-mal)-glu	0.078	0.081	0.087	0.084	0.187	0.230	0.171	0.242	0.225	0.192	0.184	0.155	1.321	2.403
Peo-3-*O*-rut	0.116	0.114	0.117	0.130	0.419	0.453	0.329	0.538	0.477	0.473	0.425	0.241	1.274	3.524
DHK	1.351	1.511	1.464	1.558	4.779	4.713	5.197	4.371	3.935	4.368	3.504	2.303	1.213	2.818

The unit was ug/g, and all values were accurate to 3 decimal places. WT1-WT4: biological replicates of wild-type K326 control tobacco; W52-W60: *PgF3′5′H* T_0_ transgenic generation tobacco lines; Pel-3-*O*-glu: pelargonidin-3-*O*-glucoside; Pel-3-*O*-rut: pelargonidin-3-*O*-rutinoside; Pel-3-*O*-(6-*O*-mal)-glu: pelargonidin-3-*O*-(6-*O*-malonyl-*β*-D-glucoside); Peo-3-*O*-rut: peonidin-3-*O*-rutinoside; DHK: dihydrokaempferol.

## Data Availability

Not applicable.

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
