# Peer review of "Heterologous Expression of Platycodon grandiflorus PgF3′5′H Modifies Flower Color Pigmentation in Tobacco"

_genes, 2023, doi:10.3390/genes14101920_

Round 1

Reviewer 1 Report

In this research, flower color pigmentation in Tobaco was modified by heterologous expression of PgF3’5’H. This manuscript is well-written and presented. However, before accepting this paper following issues should be resolved.

Line 172: “13 independent positive ….”, Cannot start a sentence with a number.

Line 311: No space between “eriodictyol” and “,”.

Line 312: No space between “DHQ” and “,”.

Line 114-115: Provide greenhouse conditions during growing plants such as light/dark cycle, temperature, and relative humidity).

Figure 2: Please add a scale bar for each panel.

In Figure 3, the gene expression data looks weird. The petal color looks similar among T0 transgenic tobacco plants, but the relative expression of PgF3’5’H in W-58 was much higher than that of other transgenic plants. Therefore, the authors should provide data on the measurement of flower color  (L*a*b* C*) and petal cell sap pH to strongly support the results.

And, what stage of the flower development was used for gene expression analysis? I think the authors should indicate this information in the material and method section. Based on the figure, it seems that the authors used flowers at the senescence stage for gene expression analysis because the discoloration (bluing) symptom appeared in the petals. 

Line 176-177: “…. and violet during withering.” Wilting of petals is one of the senescence symptoms in flowers, thus please replace “violet during withering” with “bluing during flower senescence period”.

Line 192: please replace “withering period” with “flower senescence period”.

Reviewer 2 Report

Ma et al. have transferred the putative Platycodon grandiflorus F3’5’H encoding cDNA to Nicotiana tabacum under the 35S promoter and observed a color change in tobacco flowers (from pink to purple) along with delphinidin derived anthocyanins that do not occur in nontransformed tobacco. Basically, this is a successful in vivo assay for the earlier isolated PgF3’5’H candidate gene. This, as such, is an important result since genes are not always what similarity in sequence to other genes suggests.

 The experiments are well performed and documented but the text is written in a style that implies more novelty in the results than there really is – basically all observations reported have been done before for other F3’5’H genes, including the interesting fact that anthocyanin production seems to be upregulated by ectopic F3’5’H expression. There is a missed opportunity in the work: The authors generated empty vector transformants but then used nontransgenic plants as control. The vector only transformants, or regeneration from tissue culture, could influence flavonoid biosynthesis (they are also stress induced compounds). Or if not, that would also be an interesting result.

 The authors used qPCR to determine steady state mRNA levels for the PgF3’5’H transgene. They certainly share the view that nontarnsformed tobacco does not have in the background any PgF3’5’H mRNA molecules – the background is by definition zero. Therefore, it makes no sense to compare the expression level to the background. The problem arises from the fact that qPCR gives a “value” also for zero, but this value has no meaning. The LC-MS software they use reports “Not detected” (abbreviated e.g. N.d., not N/A) when a compound is below detection level. (In this case N.d. does not even mean necessarily zero, a promiscuous hydroxylase could in principle make some molecules of delphinidin.) The authors should set a (gene specific) threshold also for qPCR. For the same reason, the calculated fold change of “1700” cannot be compared to “1500” fold found in another study – the background value depends on the gene analyzed, primers used, and on the instrument and software used. On the same line, you cannot write the PgF3’5’H was “barely” expressed in nontransgenic tobacco – it is not expressed at all (naturally).

 The authors enter the discussion section by giving the impression that they expected to see blue flowers. They discuss this well a bit further down the text, so knowing the literature they should not have expected blue color. Role of acylation is not discussed, it diverts the color strongly towards blue.

 Note that pelargonidin, cyanidin etc are not anthocyanins (they are anthocyanidins, the aglycones), so the sentence in the first paragraph of Introduction must be fixed (Anthocyanins include…).

 Third paragraph in the Introduction states that F3’5’H catalyzed hydroxylation of DHM. DHM is the product and the substrate is DHK (or DHQ) at this level.

 In Figure 2, replace “periods” by “stages”.

Some wrong/unusual terms: Fig2 "periods" should be "stages", "Not detected" should be abbreviated N.d. 

Round 2

Reviewer 1 Report

Accept in present form.